# Abnormal Expression of Prolyl Oligopeptidase (POP) and Its Catalytic Products Ac-SDKP Contributes to the Ovarian Fibrosis Change in Polycystic Ovary Syndrome (PCOS) Mice

**DOI:** 10.3390/biomedicines11071927

**Published:** 2023-07-07

**Authors:** Suo Han, Shimeng Wang, Xiang Fan, Mengchi Chen, Xiaojie Wang, Yingtong Huang, Hongdan Zhang, Yinyin Ma, Jing Wang, Chunping Zhang

**Affiliations:** 1Department of Cell Biology, College of Medicine, Nanchang University, Nanchang 330006, China; hansuo2023@163.com (S.H.); wangshim18@163.com (S.W.); 18309471993@163.com (X.F.); chenmc0323@126.com (M.C.); 15137921368@163.com (X.W.); huangyingtong21@163.com (Y.H.); 18375750882@163.com (H.Z.); 17339366776@163.com (Y.M.); 2Center for Drug Inspection of Guizhou Medical Products Administration, Guizhou Medical Products Administration, Guiyang 550081, China; 3Department of Microbiology, College of Medicine, Nanchang University, Nanchang 330006, China

**Keywords:** polycystic ovary syndrome (PCOS), prolyl oligopeptidase (POP), N-acetyl-seryl-aspartyl-lysyl-proline (Ac-SDKP), fibrosis, TGF-β1, MMP2

## Abstract

Polycystic ovary syndrome (PCOS) is an endocrine disorder and metabolic syndrome. Ovarian fibrosis pathological change in PCOS has gradually attracted people’s attention. In this study, we constructed a PCOS mouse model through the use of dehydroepiandrosterone. Sirius red staining showed that the ovarian tissues in PCOS mice had obvious fibrosis. Prolyl oligopeptidase (POP) is a serine protease and N-acetyl-seryl-aspartyl-lysyl-proline (Ac-SDKP) is its catalytic product. Studies show that abnormal expression and activity of POP and Ac-SDKP are closely related to tissue fibrosis. It was found that the expression of POP and Ac-SDKP was decreased in the ovaries of PCOS mice. Further studies showed that POP and Ac-SDKP promoted the expression of matrix metalloproteinases 2 (MMP-2) expression and decreased the expression of transforming growth factor beta 1 (TGF-β1) in granulosa cells. Hyperandrogenemia is a typical symptom of PCOS. We found that testosterone induced the low expression of POP and MMP2 and high expression of TGF-β1 in granulosa cells. POP overexpression and Ac-SDKP treatment inhibited the effect of testosterone on TGF-β1 and MMP2 in vitro and inhibited ovarian fibrosis in the PCOS mouse model. In conclusion, PCOS ovarian tissue showed obvious fibrosis. Low expression of POP and Ac-SDKP and changes in fibrotic factors contribute to the ovarian pathological fibrosis induced by androgen.

## 1. Introduction

The ovary is the reproductive organ of female animals. Its main function is to secrete sex hormones and produce mature eggs. The follicle is the basic functional unit of the ovary. According to different stages of development, follicles can be divided into primordial follicles, primary follicles, secondary follicles, antral follicles and pre-ovulatory follicles. After puberty, ovarian follicle development shows periodic changes under the action of the hypothalamic-pituitary-ovarian axis [1]. During the ovarian cycle, follicles grow in an external environment containing extracellular matrix (ECM) such as collagen, laminin, and fibronectin [2,3,4,5]. ECM not only provides structural support for developing follicles, but also regulates the growth of granulosa cells and oocytes by combining growth factors and hormones [6,7]. During ovulation, the luteinizing hormone peak induces oocyte maturation, cumulus cell expansion, follicular rupture and luteal formation. During oocyte maturation, cumulus cells express hyaluronic acid synthase 2 and secrete hyaluronic acid. This mucinous elastic matrix accumulates between cumulus cells and causes the separation and diffusion of cumulus cells, which is a key step for the maturation and ovulation of oocytes. Dysfunction of the synthesis of the cumulus matrix components can lead to reduced fertility or infertility [8]. So, ECM remodeling plays an important role in ovulation. Therefore, the normal development, maturation, and ovulation of follicles depend on the periodic degradation and remodeling of ECM.

Ovarian fibrosis, characterized by the excessive proliferation of ovarian fibroblasts and ECM deposition, is one of the main causes of ovarian dysfunction [9,10,11]. Studies have shown that ovarian fibrosis can promote the occurrence of polycystic ovary syndrome (PCOS) and premature ovarian failure (POF). In an androgen-induced PCOS model, the ovaries showed severe fibrosis [12,13]. Studies have found that the expression of MMPs and TIMPs in PCOS is also unbalanced, which leads to the abnormal deposition of ECM components such as collagen, resulting in ovulation disorders and polycystic changes in ovaries [14,15,16]. In addition, CTGF and PPAR-γ extracellular matrix regulators were abnormally expressed in PCOS patients [17]. These studies suggest that ovarian fibrosis is related to the pathological changes of PCOS.

Prolyl oligopeptidase (POP) is a unique enzyme in the serine protease superfamily that is widely expressed in mammalian tissues and organs, such as brain, ovary, liver, and testis [18]. The enzymatic activity of POP is mainly involved in the metabolism of functional polypeptides, such as vasopressin, substance P, and thyroid stimulating hormone releasing hormone, through the hydrolysis of prolyl (Pro)-XAA peptide bond (X is any amino acid except Pro) [19,20]. Its function has been extensively studied in the central nervous system. POP is mainly associated with learning and cognition, and abnormal expression and activity of POP are closely associated with Alzheimer’s disease and Parkinson’s disease [21,22]. POP also has high activity in the liver. It has been reported that circulating POP activity is significantly decreased in patients with multiple sclerosis (MS) and cirrhotic patients and in a rat model [23,24,25]. Studies showed that POP attenuated the activation of HSCs through the inhibition of TGF-β signaling and the induction of PPAR-γ and have anti-fibrosis potential in the liver [26]. A decrease in POP also contributes to the development of fibrosis in progressive nephropathy [27]. Studies also showed that POP produces N-acetyl-Seryl-aspartyl-Lysyl-proline (Ac-SDKP) by participating in the cleavage of thymosin β4. The thymosin β4-POP-Ac-SDKP axis has anti-fibrotic properties in the liver and kidneys [28,29]. These studies suggest that POP and its lysis product Ac-SDKP play an important role in inhibiting fibrosis, and their abnormal expression and activity are closely related to tissue fibrosis.

In mammalian ovaries, POP has been found to be expressed in oocytes, granulosa cells, and theca cells [18,30,31]. During estrus, POP activity has been found to be significantly increased in the ovaries [32]. In a previous study, we reported that POP is highly expressed in murine luteal cells. POP also regulates progesterone synthesis in luteal cells through the extracellular signal-regulated kinase (ERK) signaling pathway [33]. Considering the role of POP in tissue fibrosis, we detected the expression of POP in the ovaries of PCOS mice and found that POP was lowly expressed. In this study, we will explore the possible mechanism of POP on PCOS ovarian fibrosis in order to provide a new theoretical and experimental basis for understanding the mechanism of PCOS ovarian fibrosis.

## 2. Materials and Methods

### 2.1. Animals

Female Kunming mice (21 days) were purchased from the Animal Facility of Nanchang University. The mice were randomly divided into the following two groups: control group and PCOS group (*n* = 15). The control group mice were injected subcutaneously daily with 0.2 mL olive oil. The PCOS group received subcutaneous injection of DHEA (60 mg/kg body weight dissolved in 0.2 mL of olive oil) for 21 days [34]. All mice were housed in a temperature- and light-controlled facility with free access to water and food. Throughout the whole treatment period, the animals were weighed every third day. Vaginal smears were taken daily from the 1st to the 21st day of treatment. The estrous cycle stage was determined by a microscopic examination of the major cell types in vaginal smears following Wright–Giemsa staining. The diestrus stage corresponded to the stage of luteal degeneration, with a large number of white blood cells and a few epithelial cells. The control and PCOS groups were sacrificed at the diestrus stage of the estrous cycle. Blood samples were centrifuged at 1000× *g* for 10 min and the serum was collected. Concentrations of follicle stimulating hormone (FSH) and luteinizing hormone (LH) were assessed by a commercial laboratory (Beijing Sinouk Institute of Biological Technology, Beijing, China).

In order to observe the in vivo effect of POP overexpression on ovarian fibrosis in a PCOS model, we subcapsularly injected concentrated retrovirus into the ovaries of PCOS mice. In detail, 20 μL of concentrated pMIG-POP retrovirus was injected subcapsularly into the left ovary with a 10 μL syringe, and 20 μL of concentrated pMIG retrovirus was injected subcapsularly into the right ovary of the same mice. From the day after surgery, the mice received subcutaneous injection of DHEA (60 mg/kg body weight dissolved in 0.2 mL of olive oil) for 21 days. 

In order to observe the in vivo effects of Ac-SDKP on ovarian fibrosis in a PCOS model, the mice were randomly divided into two groups: a PCOS group and a PCOS+ Ac-SDKP group (*n* = 6). The mice in the PCOS group received 60 mg/kg DHEA dissolved in 0.2 mL of olive oil and 100 μL saline for 21 days. The mice in the PCOS+ Ac-SDKP group received 60 mg/kg DHEA and 800 μg/kg Ac-SDKP for 21 days. All animal experiments were approved by the Ethical Review Committee and Laboratory Animal Welfare Committee of Nanchang University (NCULAE-202209280023).

### 2.2. Hematoxylin-Eosin (HE) Staining

The ovaries were isolated and immediately fixed with 4% paraformaldehyde for 24 h. The tissues were dehydrated and embedded in paraffin. Sections of 5 μm thickness were stained with hematoxylin for 2 min and eosin for 10 s. After dehydration and transparency, the sections were sealed with neutral gum.

### 2.3. Immunohistochemistry

After deparaffinization and rehydration through degraded ethanol, the slides underwent antigen retrieval in 10 mM sodium citrate buffer for 20 min. The sections were inactivated through 3% H_2_O_2_ for 10 min, incubated with 3% BSA to block nonspecific binding, and were incubated with primary Prolyl oligopeptidase antibody (dilution 1:300, BS60084, BIOWORLD, Dublin, OH, USA) at 4 °C overnight. After washing with phosphate-buffered saline (PBS), the secondary antibody was incubated for 30 min at room temperature, and the color was developed with 3,3-diaminobenzidine for 2 min. The sections were counterstained with hematoxylin for 30 s and sealed with neutral gum.

### 2.4. Sirius Red Staining

According to Sirius Red Staining Kit instructions, the 5 μm sections were deparaffinized, rehydrated, and stained with Sirius red staining solution for 1 h. The sections were re-immersed in alcohol and xylene. The sections were sealed with neutral gum.

### 2.5. Plasmids Construction and Virus Packaging

In order to o construct POP overexpression plasmid, the whole sequence of POP was amplified by POP forward primer 5′-CCGCTCGAG ATGCTGTCCTTCCAGTACCC-3′ and POP reverse primer 5′-CCGGAATTC TTACTGGATCCACTCGATGTT-3′, digested with XhoI and EcoRI and cloned into pMig plasmid vector. DNA sequencing was used to confirm successful insertion.

In order to produce retrovirus, 293T cells were seeded in 10 cm dishes and transfected with a mixture of DNA containing 10 μg of pCL-Eco (IMGENEX, San Diego, CA, USA), 10 μg of pMIG vector, or pMIG-POP expression vectors by lipofectamine 3000 reagent. The media containing retroviruses were harvested 24 h after transfection and filtered through a 0.45 μm pore-size filter. To concentrate retrovirus, the filtered media mixed with virus precipitation solution at a volume ratio of 4:1. The mixture was gently inverted and left for 45 min at 4 °C. After being centrifuged at 7000× *g* for 45 min at 4 °C, the supernatant was completely removed, and the white precipitate was resuspended in 200 μL PBS.

### 2.6. Granulosa Cell Culture and Transfection

21-day-old Kunming female mice were injected intraperitoneally with 5 IU of pregnant mare serum gonadotropin (PMSG) to induce follicle development. After 48 h, they were sacrificed by cervical dislocation, and the ovaries were removed aseptically and washed three times with PBS. The surrounding fat and connective tissue were removed, and the ovaries were punctured with a 25-gauge injection needle under a stereoscope in order to release the granulosa cells. Cells were collected by centrifugation at 1500 rpm/min for 5 min, resuspended in F12/DMEM medium, and seeded into different culture plates according to experimental needs.

For transfection, 2 µg plasmids were transfected into granulosa cells using FUGENE-6, when cell confluence reached 70–80%. After 8 h, fresh medium was changed. The cells were treated with different reagents for 48 h and were lysed for RNA and protein extraction after.

### 2.7. Quantitative Real-Time PCR

Total RNA was extracted using TRIzol reagent and following the instructions (Invitrogen, Waltham, MA, USA), and cDNA was synthesized using a reverse transcription kit (Trans, Beijing, China). Quantitative real-time PCR was carried out at a 20 μL reaction volume, including 10 μL 2X Brilliant SYBR Green qPCR Master Mix, 1 μL cDNA, 0.5 μL primers, and 8.5 μL H_2_O. The relative expressions of each gene were determined and normalized to the expression of the housekeeping gene glyceraldehyde 3-phosphate dehydrogenase (*GAPDH*) and calculated using the 2^−ΔΔCT^ method. Primers were listed in Table 1.

### 2.8. Western Blot

Total proteins were extracted using RIPA lysis buffer. Western blot was used to detect the expression of POP, MMP2, β-actin, and TGF-β1. The total protein was separated on 12% sodium dodecyl sulfate–polyacrylamide gel electrophoresis gels and transferred to the PVDF membrane. The blot was blocked with 5% skim milk solution for 2 h at room temperature and was incubated with primary POP antibody (dilution 1: 1000, BS60084, BIOWORLD), TGF-β1 (dilution 1:500, BA2120, BOSTER, Wuhan, China), MMP2 (dilution 1:1000, BA2120, BOSTER), and β-actin (dilution 1:5000, 66009-1-Ig, Proteintech, Rosemont, IL, USA) overnight at 4 °C. After washing three times with TBST, the blot was incubated with anti-mouse IgG (dilution 1:20,000, BS12478, BIOWORLD) or anti-rabbit IgG (dilution 1:20,000, BS13278, BIOWORLD) for 1 h at room temperature. An Enhanced Easy See Western Blot Kit was employed in order to visualize the target bands, and the intensity of bands was quantified by Bio-Rad Image Laboratory software (Version 6.0). β-actin was used as an internal reference for detecting relative expression levels.

### 2.9. Ac-SDKP Measurement

Ac-SDKP levels in the ovary were detected with an Enzyme-Linked Immunosorbent Assay (ELISA) kit (MyBiosource, San Diego, CA, USA). The mice were sacrificed by cervical dislocation, and the ovaries were removed aseptically and washed with PBS. The surrounding fat and connective tissue were removed. After weighing, the ovaries were homogenized in PBS (tissue weight (g): PBS (mL) volume = 1:9). The homogenates were then centrifuged for 5 min at 5000× *g* to obtain the supernatant. The Ac-SDKP levels in the sample were determined by following the manufacturer’s protocol.

### 2.10. Statistical Analysis

All data were statistically analyzed using GraphPad Prism 7.00. The data were shown in the form of the mean and standard error of the mean (SEM). Statistical comparison between the two groups was performed by independent sample *t*-test. One-way analysis of variance followed by the Student–Newman–Keuls test was used for statistical comparisons among multiple groups. *p* < 0.05 was considered statistically significant.

## 3. Results

### 3.1. PCOS Mouse Model Was Successfully Constructed

In this study, the PCOS mouse model was constructed by subcutaneous injection of DHEA for 21 days. HE staining showed that the ovaries of mice in the control group contained various stages of growing follicles and corpus luteum. The follicular cysts in the PCOS group were obvious; the granular cell layer of follicle was reduced, and the number of corpus luteum in the ovary was reduced compared with the control group (Figure 1D). The body weight of mice in the PCOS group was increased significantly compared with the control group (Figure 1A). Hormone level detection showed that the LH and FSH levels in the PCOS group were lower than those in the control group, while LH/FSH levels were increased (Figure 1B). The estrous cycle showed that mice in the control group had regular estrous cycles, while mice in the PCOS group were stopped in the diestrus phase (Figure 1C). These results were consistent with the characteristics of PCOS, indicating that the PCOS mouse model was successfully constructed.

### 3.2. Ovarian Fibrosis Was Increased in PCOS Group

Collagen is the main component of ECM, which can combine with the highly acidic dye Sirius red, so Sirius red staining is often used to detect tissue fibrosis changes [35]. The ovarian tissues were stained with Sirius red, and the results showed that the ECM of the PCOS group was significantly thicker than that of the control group (Figure 2A). TGF-β1 is an important cytokine that promotes organ fibrosis [36]. Matrix metalloproteinases (MMPs)/tissue inhibitors of metalloproteinases (TIMPs) also play important roles in the process of tissue fibrosis [37]. We detected the expression changes of these fibrosis-related factors through real-time quantitative PCR and Western blotting and found that the mRNA expression levels of MMP-2 in the PCOS group were decreased, while the mRNA expression levels of TGF-β1 were increased. The mRNA expression levels of TIMP1 and MMP-9 were not significantly changed compared with the control group (Figure 2B). Western blotting also confirmed that TGF-β1 protein expression was increased in the PCOS group, while MMP-2 protein expression was decreased (Figure 2C). The results of the morphology and expression of fibrosis factors showed that ovarian fibrosis was significantly increased in PCOS mice.

### 3.3. POP and Ac-SDKP Were Decreased in Ovaries of PCOS Mice

It has been reported that POP attenuates the activation of hepatic stellate cells by inhibiting TGF-β1 signal transduction and inducing peroxisome proliferator-activated rector-gamma (PPAR-γ) [26]. Chronic infusion of POP inhibitors increases renal medullary fibrosis [38]. We also reported the expression of POP in mouse ovaries [33]. We detected the expression of POP through real-time PCR and Western blotting and found that the expression of POP in the PCOS group was significantly lower than that in the control group (Figure 3A,B). Immunohistochemical staining showed that POP was expressed in ovarian granulosa cells and corpus luteum, and the staining of POP in the PCOS group was significantly weaker than that in the control group (Figure 3C). As a metabolite of POP, Ac-SDKP inhibits collagen synthesis, participates in the regulation of extracellular matrix deposition and fibrosis in tissues and organs and has a significant inhibitory effect on the injury and fibrosis of the heart, liver, lungs, kidneys, and other organs [39]. We also examined the production of Ac-SDKP in ovaries and found that the concentration of Ac-SDKP in the PCOS group was decreased compared with the control group, which is consistent with the change of POP expression (Figure 3D).

### 3.4. POP and Ac-SDKP Promoted the Expression of MMP-2 Expression and Decreased the Expression of TGF-β1 in Granulosa Cells

In order to further investigate whether the decreased expression of POP and Ac-SDKP in the PCOS model mediates the changes of ovarian fibrosis, we treated primary granulosa cells with S-17092, an inhibitor of POP, and found that S-17092 inhibited the expression of MMP-2 and promoted the expression of TGF-β1. There was no significant change in MMP9 and TIMP1 (Figure 4A,B). After overexpression of POP, MMP2 expression was increased, while TGF-β1 expression was decreased. These results suggest that POP regulates the expression of TGF-β1 and MMP2 in ovarian granulosa cells (Figure 4C,D). After treatment with Ac-SDKP, MMP2 expression was also increased, while TGF-β1 expression was decreased. These results suggest that Ac-SDKP also regulates the expression of TGF-β1 and MMP2 in ovarian granulosa cells (Figure 4E,F).

### 3.5. POP and Ac-SDKP Mediated the Effect of Testosterone on the Expression of Fibrosis Factors in Granulosa Cells

Hyperandrogenemia is a typical symptom of PCOS [40], and androgen has also been widely used to induce PCOS in animal models [41]. In this experiment, the primary granulosa cells were treated with testosterone in order to observe the effect of testosterone on the expression of fibrosis factors and POP in granulosa cells. The results showed that testosterone promoted the expression of TGF-β1 and TIMP1 and inhibited the expression of MMP-2. There was no significant influence on MMP-9. Protein levels also confirmed that testosterone inhibited MMP-2 expression and promoted TGF-β1 expression. Meanwhile, real-time PCR and Western blot assays showed that testosterone inhibited the expression of POP (Figure 5A,B). The results also showed the effect of overexpression of POP and Ac-SDKP on the expression of MMP2 and TGF-β1 induced by testosterone. POP and Ac-SDKP were found to inhibit the effect of testosterone on MMP2 and TGF-β1 (Figure 5C,D).

### 3.6. POP Overexpression and Ac-SDKP Decreased the Ovarian Fibrosis in PCOS Mice

In order to further observe the effect of POP and Ac-SDKP in ovarian fibrosis in a PCOS model, POP overexpression retrovirus was injected into the ovaries, and it was found that POP overexpression reversed the pathological changes of ovarian fibrosis (Figure 6A). Ac-SDKP also rescued the fibrosis phenotype of PCOS (Figure 6B).

## 4. Discussion

PCOS is an endocrine disorder and metabolic syndrome caused by both heredity and environment [42,43]. In addition to typical polycystic change, fibrosis, as a pathological change, has gradually attracted people’s attention [9,12,44]. In this study, the PCOS model was constructed by DHEA induction, and Sirius red staining showed that PCOS mice had obvious fibrosis in the ovarian tissues compared with control mice.

TGF-β1 is a multidirectional regulatory cytokine which plays important roles in regulating cell growth, differentiation, and immune function [45]. TGF-β1 is also an important promoting organ fibrosis cytokine, which can promote the expression of ECM and inhibit the degradation of ECM. TGF-β1 inhibits the expression and activation of MMPs, up-regulates the expression of protease inhibitors such as TIMPs, and promotes the synthesis of ECM components such as type I, II, and IV collagen and fibronectin through the autocrine and paracrine pathways [46,47,48]. TGF-β1 antibody alleviates injury response and fibrosis by inhibiting the TGF-β1 signaling pathway, suggesting that blocking TGF-β1 signaling may be an effective way to prevent and treat fibrosis [49]. In the ovary, TGF-β1 is expressed in granulosa cells, theca cells, and oocytes, and plays important roles in follicle development through autocrine and paracrine pathways, including regulation of steroid hormone synthesis, ECM remodeling, and COC expansion [50]. Serum TGF-β1 levels were significantly higher in PCOS patients than in the control group [51]. TGF-β1 also plays an important role in the deposition of extracellular matrix in chocolate cysts [52,53]. In the testosterone-induced PCOS model, TGF-β mediates ovarian fibrosis by regulating the expression of fibrosis factors [13,54]. The result also showed that there was significantly increased ovarian TGF-β1 expression in the PCOS model.

MMPs are a class of highly conserved zinc ion-dependent proteolytic enzymes that play important roles in various protein degradation processes and function in tissue ECM remodeling. They are expressed in a variety of tissues and cells, but their expression level and activity are low. When the tissues are stimulated or in a pathological state, the expression of MMPs will be increased, and the activity is activated. Activated MMPs can degrade a variety of ECM components including collagen, laminin, and fibronectin. TIMPs are specific inhibitors of MMPs. The dynamic balance of MMPs/TIMPs plays an important role in tissue remodeling and injury repair. Changes in TIMPs/MMPs concentration can also affect the pathogenesis of various diseases, such as PCOS, pregnancy disorders, obesity, and metabolic syndrome [37]. Studies have shown that the disruption of the MMPS/TIMPs balance in PCOS can lead to abnormal degradation of ECM components such as collagen in follicles, resulting in ovulation disorders and polycystic changes in the ovary. However, the expression and activity of these enzymes vary greatly in different studies. Gomes et al. reported that there were no significant differences in the serum level of MMP-2, MMP-8, MMP-9, and TIMP-1 between PCOS patients and healthy volunteers, but the levels of TIMP-2 in PCOS patients were decreased [14]. Lewandowski et al. reported that serum MMP-2, MMP-9, and TIMP-1 were increased in PCOS patients, while there was no difference in TIMP-2 [15]. Lahav-baratz et al. found that MMPs’ (mainly MMP-1, MMP-2, and MMP-9) activity in the follicular fluid of PCOS patients was similar to that of the control group, but the expression of TIMP-1 protein was greatly reduced [16]. Henmi T et al. induced a PCOS rat model through DHEA and found that MMP-2 expression and activity in the PCOS model group were significantly lower than in the control group [55]. We constructed a DHEA-induced PCOS mouse model and found that the expression of MMP-2 was decreased significantly while the expression of MMP-9 and TIMP-1 did not change significantly. Combined with the abnormal expression of TGF-β1 in the PCOS model, we inferred that the abnormal expression of TGF-β1 and MMP-2 may be involved in the ovarian fibrosis of PCOS.

POP is a serine protease expressed in multiple organs which promotes the release of the antifibrotic peptide Ac-SDKP from thymosin-β4 (Tβ4). POP and Ac-SDKP play important roles in the process of anti-fibrosis [26,56]. Studies showed that Ac-SDKP reversed hypertension-induced cardiac fibrosis through the down-regulation of TGF-β1 [57]. We previously reported that POP was highly expressed in granulosa cells, theca cells, and luteal cells. POP regulates progesterone synthesis in luteal cells [33]. In this study, we detected the decreased expression of POP and Ac-SDKP in the ovaries of PCOS model mice, suggesting that POP and Ac-SDKP may be involved in the ovarian fibrosis process of PCOS.

In order to further confirm the relationship between POP-Ac-SDKP and fibrosis, we treated primary ovarian granulosa cells with POP inhibitor S-17092 and found that S-17092 promoted the expression of TGF-β1 and inhibited the expression of MMP-2. After the overexpression of POP, TGF-β1 expression was decreased while MMP2 expression was increased. Ac-SDKP treatment also decreased the expression of TGF-β1 and promoted the expression of MMP2. These results suggest that low levels of POP and Ac-SDKP may mediate the progression of ovarian fibrosis by regulating the expression of TGF-β1 and MMP-2.

Hyperandrogenemia is an important clinical feature in patients with PCOS [58]. Hyperandrogen stimulates chronic ovarian inflammation, activates NLRP3 inflammasome, and further induces a series of pathological changes, including ovarian interstitial cell fibrosis [44]. After we treated primary granulosa cells with testosterone, we found that the expression of TGF-β1 was up-regulated and MMP-2 and POP expression were down-regulated. Overexpression of POP and Ac-SDKP treatment inhibited the effect of testosterone on the expression of TGF-β1 and MMP2. These results suggest that the decreased expression of POP-Ac-SDKP and the disorder of fibrosis factors induced by high testosterone are involved in the pathological process of PCOS. Studies showed that Ac-SDKP has promising value in reducing fibrosis in the heart, liver, vessels, and kidneys. Our results also showed that POP overexpression and Ac-SDKP reversed ovarian fibrosis change in the PCOS model.

In conclusion, we found that PCOS ovarian tissue showed significant fibrosis, and the expression level of POP and Ac-SDKP was reduced in PCOS ovarian tissue. POP and Ac-SDKP may mediate androgen-induced fibrosis by affecting the expression of TGF-β1 and MMP-2.

## Figures and Tables

**Figure 1 biomedicines-11-01927-f001:**
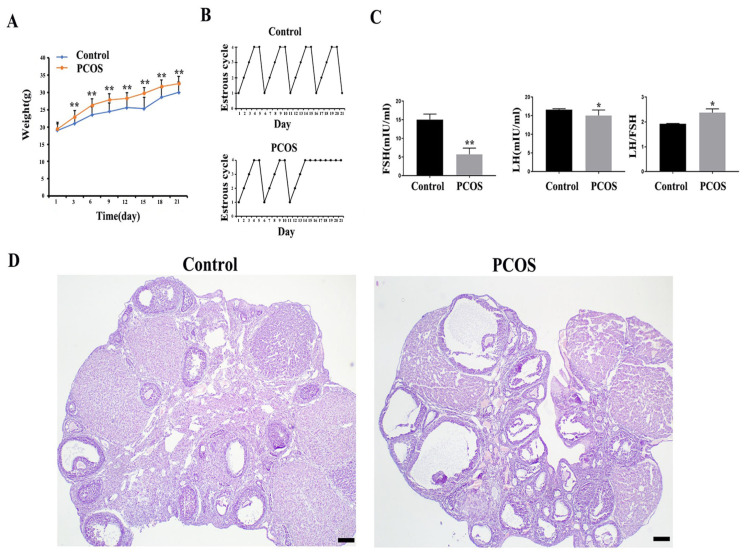
Parameters and ovarian morphological change of PCOS model mice (*n* = 15). (**A**) shows the body weight change. (**B**) shows the representative change of the estrous cycle. (**C**) shows the serum LH and FSH concentration and LH/FSH value. (**D**) shows the representative HE staining of the ovary of control and PCOS mice. * indicates *p* < 0.05. ** indicates *p* < 0.01. Scale bar = 200 μm.

**Figure 2 biomedicines-11-01927-f002:**
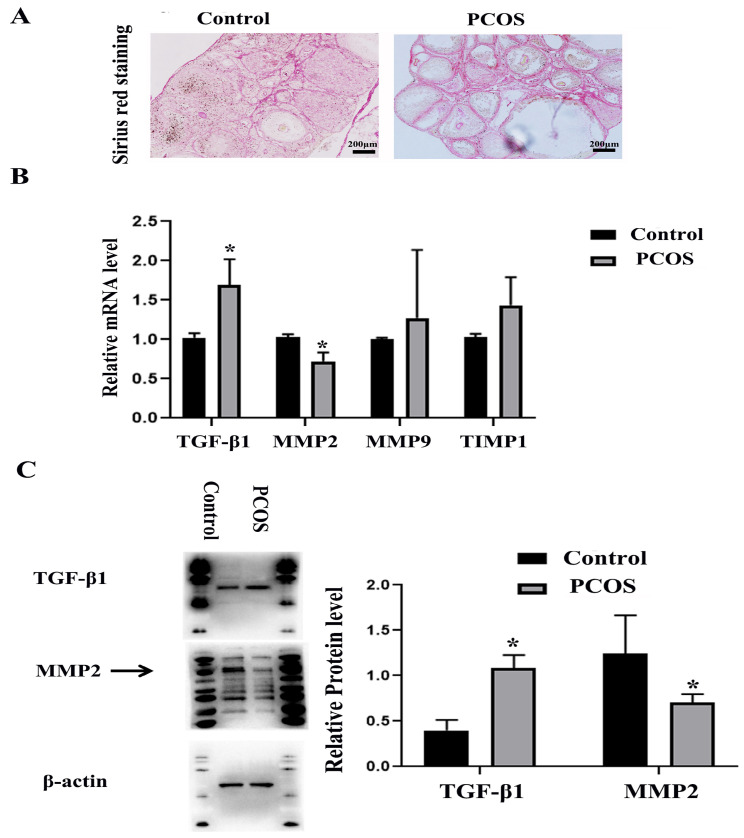
Changes of ovarian fibrosis in PCOS model mice. The changes of tissue fibrosis were detected by Sirius red staining. (**A**) shows the representative Sirius red staining of ovary in control and PCOS model mice. Scale bar = 200 μm. (**B**) shows the mRNA expression of fibrosis related factors, including MMP-2, MMP-9, TIMP1, and TGF-β1. (**C**) shows the protein expression of MMP-2 and TGF-β1. The experiment was independently repeated three times. * indicates *p* < 0.05.

**Figure 3 biomedicines-11-01927-f003:**
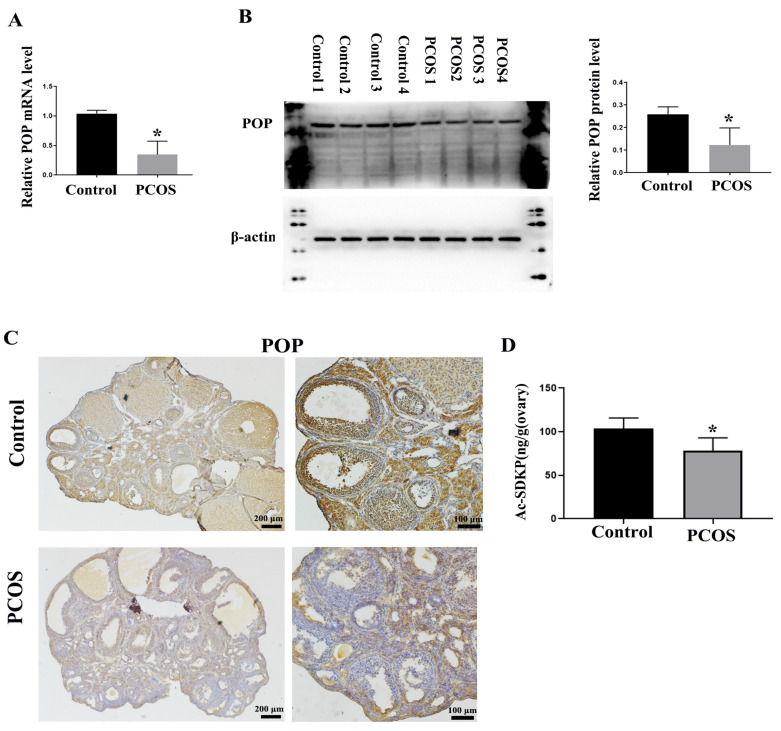
Changes in POP and Ac-SDKP expression in PCOS model mice. (**A**) shows the relative mRNA expression of POP in the ovaries of control and PCOS mice. (**B**) shows the relative protein level of POP in the ovaries of control and PCOS mice. (**C**) shows the representative immunohistochemical staining of POP in control and PCOS model mice; scale bar is indicated in the Figure. (**D**) shows the concentration of Ac-SDKP in control and PCOS model mice. * indicates *p* < 0.05.

**Figure 4 biomedicines-11-01927-f004:**
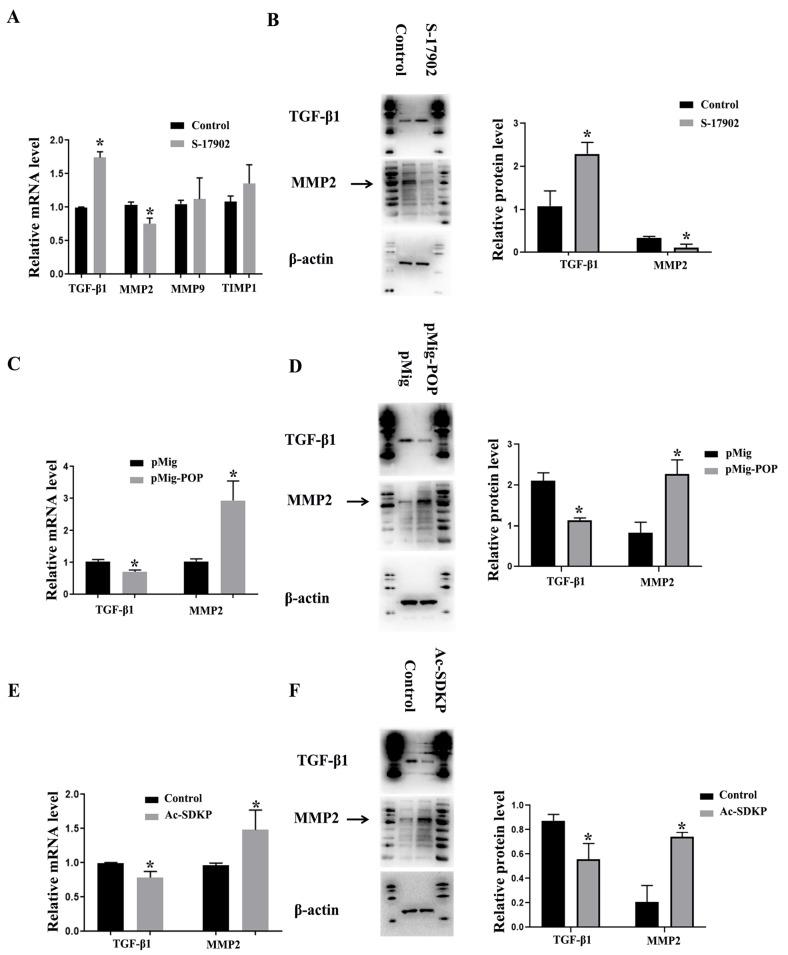
The effect of POP and Ac-SDKP on fibrosis-related factors in granulosa cells. The granulosa cells were treated with 100 µM S-17902 (POP inhibitor) or were transfected with POP overexpression plasmid for 48 h. (**A**) shows the relative mRNA expression of fibrosis-related factors, including MMP-2, MMP-9, TIMP1, and TGF-β1, after treatment with S-17902. (**B**) shows the protein expression of MMP-2 and TGF-β1 after treatment with S-17902. (**C**,**D**) show the relative mRNA and protein expression of MMP-2 and TGF-β1 after the overexpression of POP. The granulosa cells were treated with 100 nM Ac-SDKP for 48 h. (**E**,**F**) show relative mRNA and protein expression of MMP-2 and TGF-β1 after treatment with Ac-SDKP * indicates *p* < 0.05.

**Figure 5 biomedicines-11-01927-f005:**
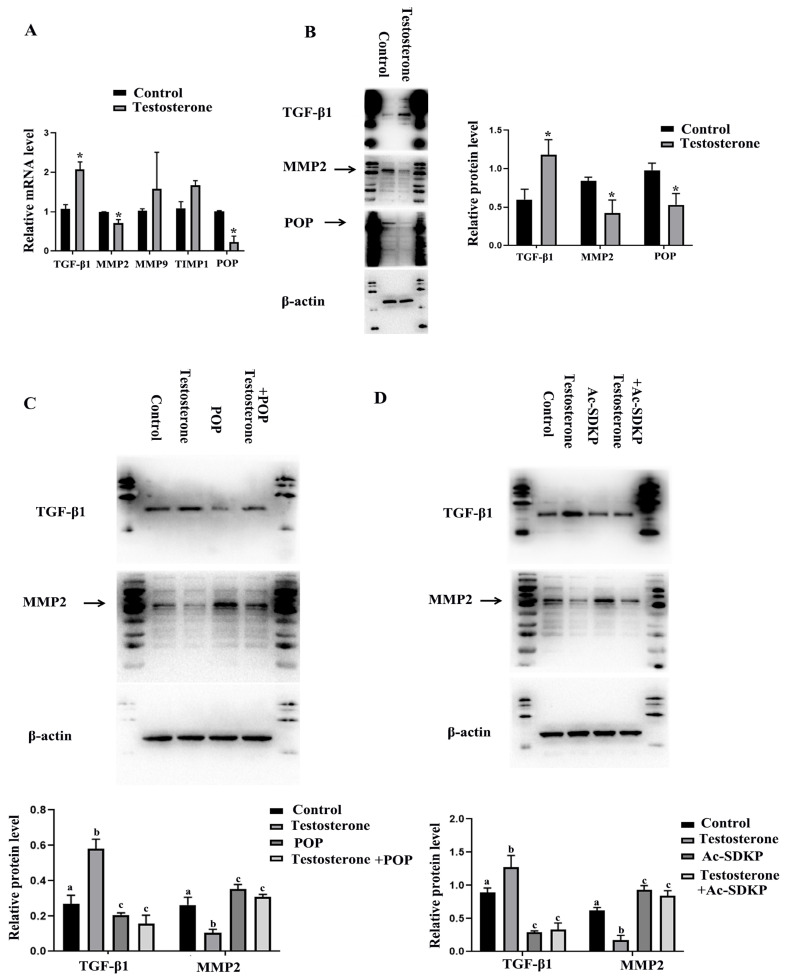
The effect of testosterone on the expression of POP and fibrosis-related factors in granulosa cells and the role of POP and Ac-SDKP in testosterone-induced fibrosis factor changes. The granulosa cells were treated with 10 µM testosterone for 48 h. (**A**) shows the mRNA expression of MMP-2, MMP-9, TIMP1, TGF-β1, and POP after treatment with testosterone. (**B**) shows the protein expression of MMP-2, TGF-β1, and POP. (**C**) shows the effect of POP overexpression on the expression of MMP-2 and TGF-β1 induced by testosterone. (**D**) shows the effect of Ac-SDKP on the expression of MMP-2 and TGF-β1 induced by testosterone. * indicates *p* < 0.05. Groups with different superscript letters are significantly different (*p* < 0.05).

**Figure 6 biomedicines-11-01927-f006:**
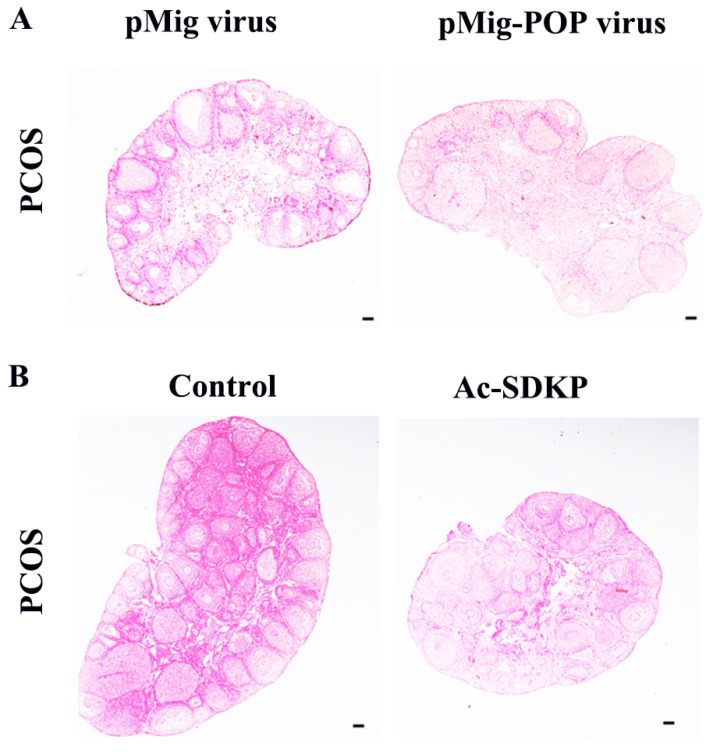
The effect of POP overexpression and Ac-SDKP treatment on ovarian fibrosis in a PCOS model. The changes of tissue fibrosis were detected by Sirius red staining. (**A**) shows the representative Sirius red staining of the ovary after injection of pMIG retrovirus and pMIG-POP retrovirus. (**B**) shows the representative Sirius red staining of the ovary after treatment with Ac-SDKP in a PCOS mice model. Scale bar = 200 μm.

**Table 1 biomedicines-11-01927-t001:** Oligonucleotides used for real-time PCR.

Gene	Sense and Antisense Primers
*POP*	5′-CCGCTCGAGATGCTGTCCTTCCAGTACCC-3′ 5′-CCGGAATTCTTACTGGATCCACTCGATGTT-3′
*TGF-β1*	5′-CCAACTATTGCTTCAGCTCCA-3′ 5′-TTATGCTGGTTGTACAGGG-3′
*MMP2*	5′-GAACACCTTCTATGGCTGC-3′ 5′-GTTGTAGTTGGCCACATCTG-3′
*MMP9*	5′-CCAAAACTACTCGGAAGACTTGC-3′ 5′-GGATACCCGTCTCCGTGCT-3′
*TIMP1*	5′-AGAACCCACCATGGCCCCCT-3′ 5′-GATTCAGGCTATCTGGGACCG-3′
*β-actin*	5′-CACGATGGAGGGGCCGGACTCATC-3′ 5′-TAAAGACCTCTATGCCAACACAGT-3′

## Data Availability

All datasets generated for this study are included in the article.

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
