# Peer review of "Abnormal Expression of Prolyl Oligopeptidase (POP) and Its Catalytic Products Ac-SDKP Contributes to the Ovarian Fibrosis Change in Polycystic Ovary Syndrome (PCOS) Mice"

_biomedicines, 2023, doi:10.3390/biomedicines11071927_

Round 1
Reviewer 1 Report
I found submitted for revision paper interesting and well-designed.
I don't have any substantial concerns to the revised manuscript.
Disappointing only are really small and difficult in reading figures in the manuscript, I would prefer their enlargement.
Reviewer 2 Report
biomedicines 2358956
“Abnormal expression of Prolyl oligopeptidase (POP) and its catalytic products Ac-SDKP contributes to the ovarian fibrosis change in Polycystic ovary syndrome (PCOS) mice”
The present work represents a significant contribution to knowledge of ovarian fibrosis induced by androgen in PCOS and hence, the knowledge of ovarian aging. The methods and techniques are detailed and sophisticated. I have some comments.
L 51: Introduction: when authors refer to “closely related”, it means that fibrosis promotes the PCOS and POF? Please clarify.
L 92: one question about the dosis used. Why was chosen that particular doses of DHA?
L 97: a comment on the staging of oestrus cycle. How was diestrus detected and which cytological features were correlated to this phase of estrus cycle?
L 108 and L 109: in order to assay the effect of Ac-SDKP in ovarian fibrosis, how many animals were allocated to each group? They belong to the first group mentioned previously (n= 15)? Please clarify.
L 112: please mention the code or reference of the ethical approval.
L 198: please try to avoid the term “we” in this part and in others in the manuscript. Rewrite the sentence using other terms, avoiding personal pronouns.
L 206: after DHEA the PCOS group should be in anestrus rather than diestrus (as diestrus is characterised by progesterone dominance, which wasn’t the case). How were the cycles staged and the phases classified?
L 219. Fig 2A. Comparing to other studies, staining of PR is weak. Any explanation? Besides, the picture does not allow/contribute significantly to the confirming of ECM fibrosis.
L 307. Also in Figure 6A the staining in not very strong. Maybe other figures are available and can be more informative in that context.
As stated above, minor editing of English language required.
